# The Association between Lymph Node Dissection and Survival in Lymph Node-Negative Upper Urinary Tract Urothelial Cancer

**DOI:** 10.3390/cancers15184660

**Published:** 2023-09-21

**Authors:** Aleksander Ślusarczyk, Piotr Zapała, Tomasz Piecha, Paweł Rajwa, Marco Moschini, Piotr Radziszewski

**Affiliations:** 1Department of General, Oncological and Functional Urology, Medical University of Warsaw, 02-005 Warsaw, Poland; 2Department of Urology, Medical University of Silesia, 41-800 Zabrze, Poland; 3Department of Urology, Comprehensive Cancer Center, Medical University of Vienna, 1090 Vienna, Austria; 4Department of Urology, Division of Experimental Oncology, Urological Research Institute, IRCCS Ospedale San Raffaele, 20132 Milan, Italy

**Keywords:** lymph node dissection, upper urinary tract urothelial cancer, radical nephroureterectomy, lymph nodes, survival, SEER

## Abstract

**Simple Summary:**

The benefit of lymph node dissection (LND) for node-negative (N0) upper urinary tract urothelial cancer (UTUC) remains uncertain. We aimed to evaluate the association between the extent of LND during radical nephroureterectomy (RNU) and survival by analyzing real-world population-based data. The removal of at least four lymph nodes was associated with improved overall and cancer-specific survival compared to no or less extensive LND. Propensity score matching was performed to adjust for confounders. Further risk-stratified subgroup analysis confirmed the survival benefit of more extensive LND, especially for muscle-invasive UTUC. Our findings underscore the significance of performing an adequate LND during RNU for N0 UTUC. Further prospective studies are crucial to confirm our results.

**Abstract:**

The benefit of lymph node dissection (LND) during radical nephroureterectomy (RNU) in lymph node (LN)-negative (cN0/pN0) UTUC remains controversial. We aimed to assess the association between LND and its extent and survival in LN-negative UTUC. The Surveillance, Epidemiology, and End Results database was searched to identify patients with non-metastatic chemotherapy-naïve cN0/pNx or pN0 UTUC who underwent RNU +/− LND between 2004 and 2019. Overall, 4649 patients with cN0/pNx or pN0 UTUC were analyzed, including 909 (19.55%) individuals who had LND. Among them, only in 368 patients (7.92%) was LND extended to at least four LNs, and the remaining 541 patients (11.64%) have had < four LNs removed. In the whole cohort, LND contributed to better cancer-specific survival (CSS) and overall survival (OS). Furthermore, a propensity score-matched analysis adjusted for confounders confirmed that improved CSS and OS was achieved only when ≥ four LNs had been removed, especially in muscle-invasive UTUC. A multivariable analysis further confirmed an association between the extent of LND and CSS. To conclude, adequate LND during RNU was associated with improved OS and CSS in LN-negative UTUC, particularly in muscle-invasive stage. This underscores that a sufficient LN yield is required to reveal a therapeutic benefit in patients undergoing RNU.

## 1. Introduction

Upper urinary tract urothelial cancer (UTUC) is a relatively rare malignancy with an unfavorable prognosis. Radical nephroureterectomy (RNU) constitutes the surgical treatment of choice in the majority of high-risk UTUC and a viable option in low-risk UTUC when kidney-sparing surgery fails or is not acceptable [1]. RNU includes the removal of the kidney with the ureter and the excision of the bladder cuff. The oncological benefit of lymph node dissection (LND) during RNU remains questionable as well-designed randomized studies are lacking [2,3]. Importantly, the additional steps of the surgery might prolong the duration of the procedure but fortunately seem not to increase the burden of perioperative complications [4]. The benefit from LND during RNU has been retrospectively demonstrated in muscle-invasive (pT2-T4) UTUC [5,6]. The advantage of LND in non-advanced and clinically node-negative (cN0) UTUC is controversial [3,7]. A systematic review of nine retrospective studies demonstrated improved cancer-specific survival (CSS) and reduced risk of local recurrence when complete and template-based LND is performed in patients with high-stage UTUC (≥pT2) [2].

Population-based studies indicate that only approximately 25–36% of RNUs include LND [4,8,9]. A proposed measure of RNU quality, the so-called tetrafecta, includes LND next to other criteria of surgical excellence such as negative surgical margins, excision of the bladder cuff, and lack of early recurrence [10]. Although many retrospective studies have shown a survival benefit associated with LND, this has not been confirmed in randomized clinical trials, and there is a lack of prospective evidence [2,5,11]. Moreover, the exact number of removed lymph nodes (LN) providing a survival benefit differs between studies and is perhaps dependent on the UTUC stage [12,13]. LND during radical cystectomy for urothelial carcinoma of the bladder has a well-established role and provides a survival advantage, which appears to be dependent on the number of removed LNs [14,15]. A similar benefit from LND is suspected in radical surgery for muscle-invasive UTUC but lacks high-quality evidence.

LND provides important staging information and enables the distinction of lymph node-positive (pN1-2) patients from lymph node-negative ones (pN0). Lymph node involvement (LNI) constitutes a strong risk factor for further recurrence and cancer-specific mortality (CSM) [3,5].

We aimed to evaluate the effect of LND and its extent on cancer-specific survival and overall survival (OS) in patients with LN-negative UTUC treated with RNU.

## 2. Materials and Methods

The National Cancer Institute Surveillance, Epidemiology, and End Results (SEER) database was utilized for this research. A search was performed to identify patients with UTUC (ICD-10 codes—C65.9-C66.9) treated surgically between 2004 and 2019. The data collected within the indicated period in 17 SEER registries covered approximately 26.5% of the United States population and were used due to their comprehensiveness and contemporariness. Patients with non-metastatic and clinically/pathologically LN-negative (cN0/pNx or pN0) UTUC, who underwent RNU (with bladder cuff excision) with or without LND were included. Patients with pN1-2 UTUC, non-primary UTUC, a previous history of bladder cancer, incomplete information on tumor T stage, an autopsy-based diagnosis, lacking survival status, and those who underwent neoadjuvant chemotherapy or radiotherapy were excluded. The available data included information on the patient’s demographics, histopathological and clinical characteristics, the therapy used, and survival outcomes.

### 2.1. Ethics

Due to the study character the institutional review board approval was not required. The study was performed in accordance with the Declaration of Helsinki and its later amendments.

### 2.2. Statistical Analysis

Categorized values were presented as percentage and number of patients. Continuous variables were presented as median accompanied by interquartile range (IQR). Frequency differences between cohorts were tested with the exact Fisher’s test or Chi-square tests whenever required. Median follow-up was computed using the reverse Kaplan–Meier method. Survival estimates were generated from Kaplan–Meier curves. Kaplan–Meier curves with 95% confidence intervals (CI) and multiple log-rank tests with Tukey–Kramer correction were used to assess the effect of LND on survival.

Propensity score matching (PSM) was performed between patients who underwent LND and those who did not, with adjustment for the following confounders: T stage, grade, tumor size, tumor location, age, and gender. Additionally, forced matching was employed to ensure that treated ones and their matched control counterparts shared identical values for tumor T stage and tumor location. The propensity scores were calculated based on logistic regression to estimate the probability of receiving treatment, and the support region was extended by the logit of the propensity score. The method employed was optimal matching with a two-to-one ratio of control to treated units in order to minimize the total within-pair difference, thereby enhancing the comparability of the study groups.

Cox proportional hazards (CPH) were used for the prediction of survival outcomes (CSS and OS). A multivariable analysis was performed to identify independent predictors of CSM in the whole cohort and in the well-balanced cohort after PSM. The stepwise variable selection method was used with a significance level of 0.2 to enter and 0.05 to stay in the multivariable model for each variable. Additionally, competing-risk regression using CPH was performed to account for other-cause mortality. Hazard ratios (HR) supplemented with a 95% CI were derived from CPH. For all statistical analyses, we considered a two-sided *p*-value < 0.05 as statistically significant. Statistical analyses were performed in SAS software version 9.4.

## 3. Results

### 3.1. Cohort Characteristics

Overall, 4649 patients with cN0/pNx or pN0 UTUC underwent RNU, including 909 (19.6%) individuals in whom LND was performed. Among them, in 368 patients (7.9%), at least four lymph nodes were dissected. In the remaining 541 (11.6%) patients, LND was limited to less than four lymph nodes.

The cohort included 2620 males (56.4%) and 2029 (43.6%) females. The majority of patients were older than 70 years old (N = 2862; 61.6%). Muscle-invasive tumors were diagnosed in 2933 (63.1%) patients and high-grade histology was reported in 3433 (73.8%) patients. Tumors located in the renal pelvicalyceal system were predominant (N = 3038; 65.3%), and the remaining originated from the ureter (N = 1611; 34.7%).

During the median study follow-up of 100 months (IQR 61-143 months), 2534 (54.5%) deaths were recorded and 1430 (30.8%) were attributable to UTUC.

Detailed information about all baseline characteristics is provided in Table 1.

### 3.2. Factors Associated with LND Performance

LND was more frequently performed in a more advanced UTUC stage (*p* = 0.002), in high-grade tumors (*p* < 0.001), in younger patients (*p* = 0.013), after the year 2010 (*p* < 0.001), and for UTUC localized in the ureter (*p* = 0.001) (Table 2). Patients who underwent LND more frequently received adjuvant chemotherapy (20% vs. 12.8%; *p* < 0.001).

### 3.3. Association between LND and Survival

In the LND group, there were 1205 (32.2%) cancer-specific deaths and 2136 (57.1%) all-cause deaths, whereas in the no LND group, there were 225 (24.8%) cancer-specific deaths and 398 (43.8%) all-cause deaths (Table 2). In the Kaplan–Meier analyses, LND performance was associated with improved CSS and OS (*p* < 0.01 and *p* < 0.001, respectively) (Figure 1A,B). The estimates of 5-year CSS (71% vs. 66.2%; *p* = 0.004) and 5-year OS (58.9% vs. 51.9%; *p* < 0.001) were better in the LND group compared to its counterpart. After further detailed analysis according to the extent of LND, we observed CSS and OS advantage in patients in whom at least 4 lymph nodes (extended LND) were removed (*p* < 0.001 and *p* < 0.001, respectively) (Figure 1C,D). Removal of < 4 LNs was not associated with CSS or OS benefit compared to no LND (*p* = 0.25 and *p* = 0.10, respectively). Extended, but not limited LND was associated with better CSS when compared to no LND in both non-muscle-invasive and muscle-invasive UTUC (Figure 1E,F).

### 3.4. Association between the Extent of LND and Survival

After PSM, the LND and no LND cohorts were adjusted for selected confounders (T stage, grade, tumor size, tumor location, age, and gender) and there were no differences in baseline characteristics between the groups (Table 3). After PSM, a risk-matched cohort of 2526 patients was analyzed and the CSS benefit from the removal of ≥ 4 LNs compared to limited LND (*p* = 0.003) or no LND (*p* < 0.001) also held true, but only in muscle-invasive UTUC (*p* = 0.029 and *p* = 0.003, respectively). Extended LND was associated with better CSS than limited LND in muscle-invasive UTUC (pT2-T4) (*p* = 0.029), but not in non-muscle-invasive TaT1 UTUC (*p* = 0.12) (Figure 2). LND limited to fewer than 4 LNs did not bring any CSS advantage compared to no LND in either subgroup (*p* > 0.05) (Figure 2).

### 3.5. Predictors of CSS in N0 UTUC

Multivariable analyses of the whole cohort and the propensity score-matched group demonstrated an independent prognostic role of LND extended to at least 4 LNs (extended LND vs. no LND; HR = 0.60 95% CI 0.46–0.78 *p* < 0.001; limited LND vs. no LND; HR = 0.94 95% CI 0.78–1.12 *p* = 0.46) in N0 UTUC treated with RNU (Table 4). Other factors associated with CSS included tumor T stage (T2 vs. T1; HR = 1.48 95% CI 1.16–1.89; T3 vs. T1; HR = 2.55 95% CI 2.08–3.13; T4 vs. T1; HR = 4.60 95% CI 3.40–6.21; *p* < 0.001), grade (HG vs. LG; HR = 2.15 95% CI 1.62–2.86; *p* < 0.001), age (60–70 years vs. <60 years; HR = 1.51 95% CI 1.13–2.02; 70–80 years vs. <60 years; HR = 1.87 95% CI; 1.42–2.47; >80 years vs. <60 years; HR = 2.71 95% CI 2.04–3.60; *p* < 0.001), UTUC location (pelvis vs. ureter; HR = 0.75 95% CI 0.64–0.87; *p* < 0.001), and tumor size (≥ 2 cm vs. < 2 cm; HR = 1.44 95% CI 1.12–1.84; *p* < 0.01). Additionally, in a separate multivariable analysis, an independent association between the number of removed LNs (HR = 0.96 95%CI 0.93–0.98 *p* < 0.01) and CSS was confirmed after adjustment for the above factors.

Furthermore, competing-risk Cox proportional hazards regression, adjusted for tumor T stage, grade, age, tumor size, and location, demonstrated an independent prognostic role of LND extended to at least 4 LNs (extended LND vs. no LND; HR = 0.64 95% CI 0.49–0.83 *p* = 0.001; limited LND vs. no LND; HR = 0.94 95% CI 0.78–1.13 *p* = 0.48).

## 4. Discussion

In this population-based study, we evaluated the effect of LND and its extent on survival in LN-negative UTUC treated with RNU. Firstly, we found that in the general cohort of N0 UTUC patients, LND during RNU was associated with advantages in CSS and OS. Secondly, after stratification based on the extent of LND, we found that the removal of at least four LNs was associated with improved CSS, whereas limited LND (removal of <four LNs) was not associated with CSS advantage compared to no LND. Thirdly, after applying PSM in stage-stratified analyses, extended LND was associated with better CSS only in muscle-invasive UTUC. The multivariable analysis confirmed the independent impact of the extent of LND on CSS. Fourthly, LND remains underutilized, even in patients with locally advanced UTUC. When performed, only a small number of LNs are typically removed. The reason for the observed survival benefit from LND during RNU for N0 UTUC might be the more adequate nodal staging and perhaps the removal of micrometastases not found on routine histopathological examinations. Evidently, patients incorrectly staged as pN0 due to a small number of examined LNs have a significantly worse prognosis than truly LN-negative ones. Abe et al. highlighted the issue of routinely undetected micrometastases which were found in 14% of pN0 UTUC [11]. Moreover, we observed that limited LND for N0 UTUC was associated with worse survival than extended LND, possibly due to the suspected under-staging when only up to three LNs were removed. The removal of a higher number of LNs, including those with routinely undetected micrometastases, might explain the observed survival benefit from LND in N0 UTUC. Conventional imaging techniques (e.g., computed tomography) are of moderate accuracy and poor sensitivity for the assessment of clinical nodal staging in urothelial cancer. Therefore, a diagnosis of cN0 disease should not preclude LND performance [16]. Furthermore, the risk of LNI in muscle-invasive UTUC is substantial [3]. Consequently, the removal of more LNs may be associated with the eradication of unnoticed LN micrometastases and a subsequent improvement in the recurrence rate and CSS, suggesting that an extended template-based lymphadenectomy may have therapeutic benefits [17].

We believe that, based on our results and previous reports, meticulous template-based LND should be the necessary step of every RNU for suspected muscle-invasive UTUC regardless of clinical nodal staging. Several studies favor the performance of LND and provide evidence of the survival benefits of that procedure during RNU [2,5,6,11]. Dong et al. showed that cN0 patients achieve better survival when LND is performed. However, the extension of LND and the number of LNs removed were not analyzed in that study [6]. Zhai et al. reported the beneficial effect of LND, especially when four or more regional LNs were removed, on survival in pT3-4 UTUC [18]. But it is worth noting that this analysis included a proportion of node-positive patients [18]. In two other papers by Roscigno et al. and Abe et al., it was not specified that all pNx patients had prior imaging and were clinically node-negative, which should be considered as a limitation of these studies [5,11]. Importantly, the benefits of LND must be contextualized with T stage as the CSS improvement is suggested only in muscle-invasive or locally advanced UTUC [2,5,7,18].

On the other hand, some retrospective studies showed that there was no survival difference between cN0 patients who did and did not undergo LND, but the subgroup analyses according to the number of removed LNs were not performed [19,20]. One of the multicenter retrospective analyses demonstrated that in cN0 muscle-invasive UTUC (≥pT2 stage) LND did not provide a survival benefit when compared to no LND [20]. The lack of benefit from LND observed in many retrospective studies might be due to the incompleteness of the dissection and the inclusion of cNx or even cN1 patients “incorrectly” classified as pNx due to the lack of histopathological examination. Furthermore, the survival benefit from LND in node-positive UTUC is also controversial and probably limited to selected patients with low burden of LN metastasis, in whom LND might be therapeutic. Xia et al. demonstrated that in patients with node-positive UTUC, removing more LNs does not offer a better therapeutic effect. However, positive LN density provided additional prognostic value for CSS and OS [21]. Another study showed that in clinically node-positive patients with UTUC, performing LND in addition to RNU at any clinical stage does not seem to have a significant impact on OS [22]. Only a meticulously designed randomized trial can definitively address the question of the benefits and optimal extent of LND in patients with UTUC. Importantly, in our study we focused only on node-negative UTUC and excluded patients with cN1 or pN1 UTUC, who clearly have a worse prognosis than LN-negative ones. Moreover, we excluded the group of cNx patients which likely contains some proportion of undiagnosed pN1 individuals, possibly contributing to the worse prognosis of this cohort.

Our observations on the oncological benefits of more extended LND strengthen the previous reports of other authors, although different cut-offs for the optimal number of removed LNs were proposed [8,12]. Roscigno et al. showed that in the entire population of UTUC the number of removed LNs was not prognostic for CSS, but in pN0 patients removal of ≥ eight LNs provided better CSS than the dissection of < eight LNs [12]. Chappidi et al. performed a SEER-based population study and demonstrated that the first quartile of patients with the highest number of removed LNs (≥five) was characterized by better CSS than other quartiles in both subgroups of pN0 and pN1-3 patients [8]. Therefore, the subgroup analyses with the inclusion of patients with more extensive LND are crucial to establish evidence for the therapeutic and prognostic role of LND during RNU.

Here, we present an analysis of a more contemporary, larger and propensity score-matched cohort of N0 UTUC patients who underwent RNU. The most important finding of our study is the fact that in N0 UTUC, LND was associated with CSS and OS benefits, but only when a sufficient number of LNs were removed. Firstly, the lack of CSS benefit in N0 patients who underwent limited LND (< four LNs removed) compared to no LND may indicate that removal of fewer than four LNs does not provide adequate staging (some patients are still under-staged). Secondly, some undetected micrometastatic LNs might still be missed during less extensive LND and contribute to nodal and distant recurrence. Perhaps therapeutic benefit might be observed only after the removal of more LNs, including micrometastatic ones which are falsely reported as pN0. A paper by Xylinas et al. showed that the number of dissected LNs should be higher with each advancement of the T stage [13]. With two dissected LNs, patients with pT0-Ta-Tis-T1 UTUC would have a greater than 95% chance of receiving the proper pathologic nodal staging. To achieve the same accuracy, in patients with pT3-T4 disease, removal of more than twelve LNs is required [13]. Another study by Roscigno et al. also underlined the importance of extensive LND to achieve correct nodal staging. In that study, it was suggested that the removal of eight LNs resulted in a 75% probability of finding at least one metastatic LN [23]. Our analysis shows that the removal of at least four LNs already provides a CSS benefit in cN0 UTUC, which stays in line with previous observations on the crucial prognostic and therapeutic role of adequate LND. We signalize the survival benefits from more accurate LND in N0 UTUC, but further investigation on the optimal LN yield in stage-stratified cohorts should be carried out. Based on available evidence and our observations, a template-based LND should be an inherent part of RNU for muscle-invasive UTUC, aiming for more adequate staging and potential eradication of regional metastasis.

In the multivariable analysis we identified several risk factors influencing CSS. Besides the extent of LND, tumor T stage and grade, tumor size and location, and patient’s age were independently associated with CSS. The most commonly considered prognostic factors in UTUC, such as tumor T stage, grade, and patient age, are strong determinants of oncological outcomes and guide clinical decision making in the postoperative oncological care. A recent randomized clinical trial demonstrated that adjuvant gemcitabine-platinum chemotherapy prolongs disease-free survival after RNU for locally advanced UTUC [24]. UTUC treatment seems to be especially challenging in the cohort of elderly people in whom adjuvant regimens are perhaps less frequently used [24]. The inadequate management of elderly individuals has also been observed in the context of urothelial bladder cancer research [25]. Several studies have reported an independent negative prognostic role of older age for CSS in various stages of urothelial cancer of the bladder and upper urinary tract [26,27,28,29]. A recent multicenter study has also shown that patients aged 70 and above undergoing RNU for UTUC had worse OS and CSS compared to younger individuals [29].

Our study has several limitations owing to its population-based nature. Most importantly, we did not receive the information on the template of LND and pathology reports were not centralized. We lacked the detailed data on clinical imaging used for preoperative nodal staging which might have influenced the clinician’s decision on LND performance. To manage this issue we restricted enrollment to cN0/pNx and pN0 patients. Patients with cNx/pNx UTUC were excluded to diminish the bias, which would have been otherwise introduced by analyzing patients who might have had cN1 UTUC and no LND performed (pNx). The modality of surgery and the multiplicity of tumors were also not reported. To overcome the residual confounding, propensity score matching and subgroup analyses were introduced. Patients who underwent neoadjuvant chemotherapy were excluded to avoid the bias of analyzing initially node-positive responders who were down-staged to ypN0 disease.

## 5. Conclusions

In conclusion, our population-based study demonstrated that adequate LND during RNU is associated with improved overall and cancer-specific survival in patients with lymph node-negative UTUC, particularly in individuals with muscle-invasive disease. However, LND remains underutilized during RNU and most often results in a low lymph node yield. LND provides essential staging information and might lead to eradication of undetected nodal micrometastases, ultimately contributing to improved oncologic outcomes. Further prospective trials are warranted to validate our findings.

## Figures and Tables

**Figure 1 cancers-15-04660-f001:**
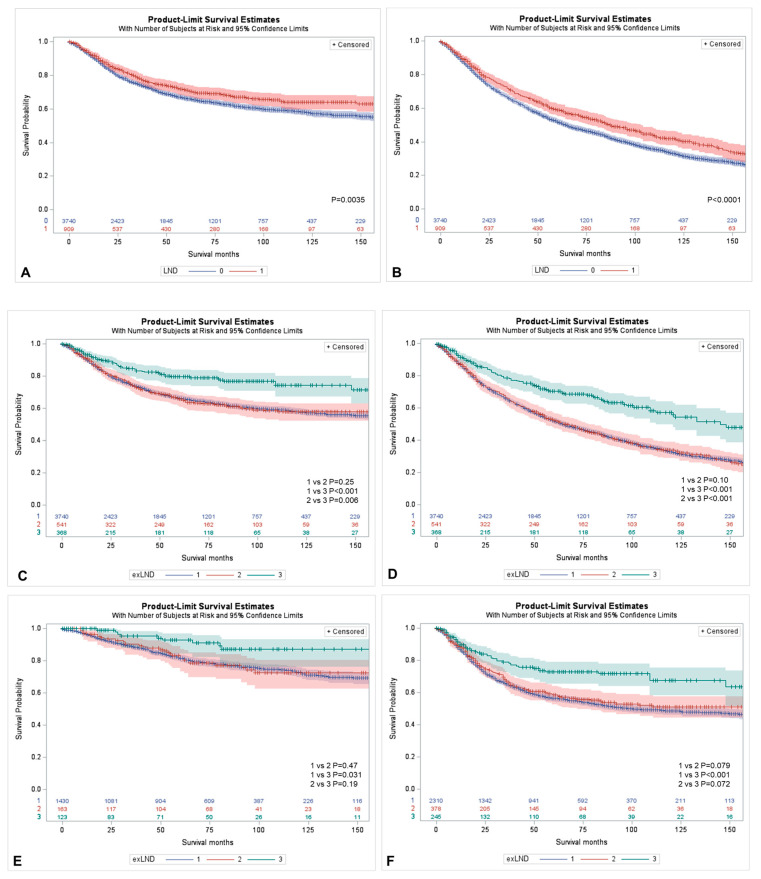
Kaplan–Meier curves for the comparison of the influence of the lymph node dissection on cancer-specific survival (CSS) and overall survival (OS) in node-negative UTUC patients undergoing RNU. (**A**) CSS for LND vs. no LND; (**B**) OS for LND vs. no LND; (**C**) CSS for removal of ≥ 4 LNs (curve 3) vs. removal of 1–3 LNs (curve 2) vs. no LND (curve 1); (**D**) OS for removal of ≥ 4 LNs (curve 3) vs. removal of 1–3 LNs (curve 2) vs. no LND (curve 1); (**E**) CSS for removal of ≥ 4 LNs (curve 3) vs. removal of 1–3 LNs (curve 2) vs. no LND (curve 1) in the subgroup of TaT1 UTUC; (**F**) CSS for removal of ≥ 4 LNs (curve 3) vs. removal of 1–3 LNs (curve 2) vs. no LND (curve 1) in the subgroup of T2-T4 UTUC.

**Figure 2 cancers-15-04660-f002:**
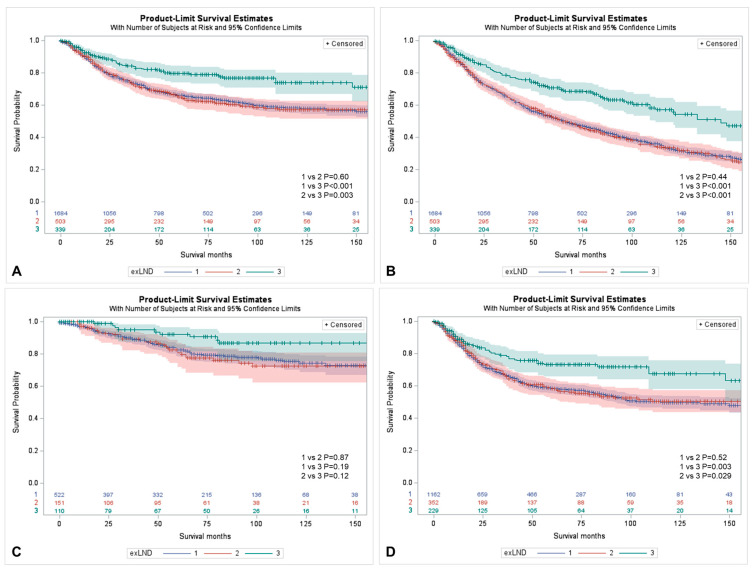
Kaplan–Meier curves for the comparison of the influence of the lymph node dissection on survival in node-negative UTUC patients undergoing RNU—cohorts after propensity score matching. (**A**) CSS for removal of ≥ 4 LNs (curve 3) vs. removal of 1–3 LNs (curve 2) vs. no LND (curve 1); (**B**) OS for removal of ≥ 4 LNs (curve 3) vs. removal of 1–3 LNs (curve 2) vs. no LND (curve 1); (**C**) CSS for removal of ≥ 4 LNs (curve 3) vs. removal of 1–3 LNs (curve 2) vs. no LND (curve 1) in the subgroup of TaT1 UTUC; (**D**) CSS for removal of ≥ 4 LNs (curve 3) vs. removal of 1–3 LNs (curve 2) vs. no LND (curve 1) in the subgroup of T2-T4 UTUC.

**Table 1 cancers-15-04660-t001:** Baseline characteristics of lymph node-negative (cN0/pNx or pN0) upper urinary tract urothelial cancer (UTUC) patients who underwent radical nephroureterectomy (RNU) with or without lymph node dissection.

Characteristics of the Whole Cohort			
Variable		Number of pts.	%
Gender	male	2620	56.36
	female	2029	43.64
Age (years)	<60	623	13.40
	60–70	1164	25.04
	70–80	1637	35.21
	>80	1225	26.35
Lymph node dissection	not performed	3740	80.45
	<4 LN removed	541	11.64
	≥4 LN removed	368	7.92
Grade	low-grade	902	19.40
	high-grade	3433	73.84
	unknown	314	6.75
Tumor T category	Ta	28	0.60
	T1	1688	36.31
	T2	916	19.70
	T3	1836	39.49
	T4	181	3.89
Tumor size	<2 cm	624	13.42
	≥2 cm	3563	76.64
	unspecified	462	9.94
Location	renal pelvis	3038	65.35
	ureter	1611	34.65
Laterality	right	2328	50.08
	left	2318	49.86
	unspecified	3	0.06
Year of diagnosis	<2010	1872	40.27
	≥2010	2777	59.73
Adjuvant chemotherapy	no/ unknown	3986	85.74
	yes	663	14.26
Race	White	4056	87.24
	Black	196	4.22
	other *	397	8.54
Marital status	single/divorced/widowed	1722	37.04
	married **	2767	59.52
	status unknown	160	3.44
Income annually	<USD 65,000	2109	45.36
	≥USD 65,000	2540	54.64
Metropolitan citizenship ***	no	622	13.38
	yes	4024	86.56
Cancer-specific death	no	3219	69.24
	yes	1430	30.76
All-cause death	no	2115	45.49
	yes	2534	54.51

* includes American Indian/Alaska Native/Asian or Pacific Islander. ** partnership without official marriage was also regarded as married status in the analysis. *** citizenship of counties in a metropolitan area. LN—lymph nodes.

**Table 2 cancers-15-04660-t002:** Comparison between the characteristics of node-negative UTUC patients who underwent lymph node dissection (LND) and those who did not.

Whole Cohort—before Propensity Score Matching						
Characteristics		No LND		LND		
Variable		Number of pts.	%	Number of pts.	%	*p*-Value
Gender	male	2118	56.63	502	55.23	0.46
	female	1622	43.37	407	44.77	
Age (years)	<60	490	13.10	133	14.63	0.013
	60–70	923	24.68	241	26.51	
	70–80	1303	34.84	334	36.74	
	>80	1024	27.38	201	22.11	
Grade	low-grade	781	22.36	121	14.37	<0.0001
	high-grade	2712	77.64	721	85.63	
Tumor T category	Ta	26	0.70	2	0.22	0.002
	T1	1404	37.54	284	31.24	
	T2	722	19.30	194	21.34	
	T3	1449	38.74	387	42.57	
	T4	139	3.72	42	4.62	
Tumour size	<2 cm	508	13.58	116	12.76	0.047
	≥2 cm	2842	75.99	721	79.32	
	unspecified	390	10.43	72	7.92	
Location	pelvis	2487	66.50	551	60.62	0.001
	ureter	1253	33.50	358	39.38	
Year of diagnosis	<2010	1595	42.65	277	30.47	<0.0001
	≥2010	2145	57.35	632	69.53	
Adjuvant chemotherapy	no/unknown	3260	87.17	726	79.87	<0.0001
	yes	480	12.83	183	20.13	
Race	White	3289	87.94	767	84.38	0.014
	Black	152	4.06	44	4.84	
	other *	299	7.99	98	10.78	
Cancer-specific death	no	2535	67.78	684	75.25	<0.0001
	yes	1205	32.22	225	24.75	
All-cause death	no	1604	42.89	511	56.22	<0.0001
	yes	2136	57.11	398	43.78	

* includes American Indian/Alaska Native/Asian or Pacific Islander.

**Table 3 cancers-15-04660-t003:** Comparison between the characteristics of propensity score-matched (PSM) cohorts of node-negative UTUC patients who did and did not undergo lymph node dissection (LND) during RNU.

Propensity Score-Matched Cohort						
Characteristics		No LND		LND		
Variable		Number of pts.	%	Number of pts.	%	*p*-Value
Gender	male	971	57.66	465	55.23	0.25
	female	713	42.34	377	44.77	
Age (years)	<60	226	13.42	116	13.78	0.95
	60–70	461	27.38	228	27.08	
	70–80	609	36.16	311	36.94	
	>80	388	23.04	187	22.21	
Grade	low-grade	239	14.19	121	14.37	0.90
	high-grade	1445	85.81	721	85.63	
Tumor T category	Ta	4	0.24	2	0.24	1.0
	T1	518	30.76	259	30.76	
	T2	358	21.26	179	21.26	
	T3	728	43.23	364	43.23	
	T4	76	4.51	38	4.51	
Tumor size	<2 cm	226	13.42	110	13.06	0.57
	≥2 cm	1356	80.52	672	79.81	
	unspecified	102	6.06	60	7.13	
Location	pelvis	1012	60.10	506	60.10	1.0
	ureter	672	39.90	336	39.90	
Year of diagnosis	<2010	528	31.35	263	31.24	0.96
	≥2010	1156	68.65	579	68.76	
Adjuvant chemotherapy	no/unknown	1402	83.25	682	81.00	0.17
	yes	282	16.75	160	19.00	
Race	White	1459	86.64	712	84.56	0.33
	Black	65	3.86	40	4.75	
	other *	160	9.50	90	10.69	
Cancer-specific death	no	1154	68.53	628	74.58	0.016
	yes	530	31.47	214	25.42	
All-cause death	no	772	45.84	468	55.58	<0.0001
	yes	912	54.16	374	44.42	

* includes American Indian/Alaska Native/Asian or Pacific Islander.

**Table 4 cancers-15-04660-t004:** Multivariable analyses using Cox proportional hazards for the prediction of cancer-specific survival (CSS) in node-negative UTUC before and after propensity score matching (PSM).

Factors Predicting Cancer-Specific Survival—Multivariable Analysis							
Variables		before PSM			after PSM		
		HR	95% CI	*p*-value	HR	95% CI	*p*-Value
Lymph node dissection	not performed	ref		<0.0001	ref		0.0008
	<4 LN removed	0.858	0.724–1.017	0.0775	0.935	0.781–1.120	0.4653
	≥4 LN removed	0.559	0.430–0.727	<0.0001	0.598	0.457–0.783	0.0002
Tumor T category	T1	ref		<0.0001	ref		<0.0001
	Ta	1.317	0.586–2.961	0.5056	1.334	0.185–9.606	0.7750
	T2	1.440	1.209–1.714	<0.0001	1.477	1.156–1.887	0.0018
	T3	2.540	2.205–2.927	<0.0001	2.553	2.084–3.128	<0.0001
	T4	5.481	4.373–6.870	<0.0001	4.598	3.404–6.211	<0.0001
Grade	high vs. low	1.782	1.511–2.101	<0.0001	2.149	1.617–2.856	<0.0001
Age (years)	<60	ref		<0.0001	ref		<0.0001
	60–70	1.420	1.142–1.766	0.0016	1.511	1.131–2.017	0.0052
	70–80	1.938	1.580–2.377	<0.0001	1.872	1.421–2.466	<0.0001
	>80	2.749	2.234–3.382	<0.0001	2.714	2.044–3.604	<0.0001
Tumor size	<2 cm	ref		0.0004	ref		
	≥2 cm	1.261	1.055–1.506	0.0108	1.438	1.122–1.843	0.0041
	unspecified	1.576	1.255–1.979	<0.0001	1.673	1.185–2.363	0.0035
Location	pelvis vs. ureter	0.754	0.671–0.848	<0.0001	0.749	0.641–0.874	0.0003
Income annually	<USD 65,000 vs. ≥USD 65,000	1.125	1.010–1.252	0.0324	-	-	-

LN—lymph nodes; HR—hazard ratio; CI—confidence interval.

## Data Availability

All data used for the analysis were from the SEER database and access to the data can be obtained from the U.S. National Cancer Institute.

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
