# Peer review of "The Association between Lymph Node Dissection and Survival in Lymph Node-Negative Upper Urinary Tract Urothelial Cancer"

_cancers, 2023, doi:10.3390/cancers15184660_

Round 1

Reviewer 1 Report

From a biostats and clinical epidemiology point of view, this manuscript has been well planned, being based on SEER data (not original ones). 

Some suggestions for the Authors:

- why have you choosen the time frame 2004-2019, is it the only one available on SEER? please, specify it in the text

- please, add to stats section that continuous covariates are reported as median/IQR

- table 1 lymph node dissection, tumor size and annual income, are they available as exact values (= continuous covariates)? If yes, it would be reccomended to add these infos and use them in your prognostic models

- all around the manuscript, p-values with 3-sign digits

- all around the manuscript, HRs and 95% CIs with 2-sign digits

- Propensity-score matched cohort, all info is lacking, please provide full detail about this procedure (how have you done it!?)

- as a further step, please describe how you have used PS matching in the Cox PH models, actually this detail is lacking

- please, add the number of OS and CSS events, globally and stratified by lymph node dissection 

minor

Author Response

From a biostats and clinical epidemiology point of view, this manuscript has been well planned, being based on SEER data (not original ones). 

Author Response: Thank you. We are honored for the kind reception of our work and greatly appreciate your suggestions for further analyses to improve our study.

Some suggestions for the Authors:

- why have you choosen the time frame 2004-2019, is it the only one available on SEER? please, specify it in the text

Author Response: Thank you for this thoughtful suggestion. This was the most contemporary and comprehensive cohort available for the analysis. We added additional information to the methods section and the paragraph is as follows: “The National Cancer Institute Surveillance, Epidemiology, and End Results (SEER) database was utilized for this research. A search was performed to identify patients with UTUC (ICD-10 codes - C65.9-C66.9) treated surgically between 2004 and 2019. The data collected within the indicated period in 17 SEER registries covered approximately 26.5% of the United States population and were used due to their comprehensiveness and contemporariness.”

- please, add to stats section that continuous covariates are reported as median/IQR

Author Response: Thank you for this thoughtful comment. We added the sentence to the methods section “Continuous variables were presented as median accompanied by interquartile range (IQR).”

- table 1 lymph node dissection, tumor size and annual income, are they available as exact values (= continuous covariates)? If yes, it would be reccomended to add these infos and use them in your prognostic models

Author Response: We are extremely grateful to the reviewer for this thoughtful comment. The exact number of lymph nodes removed was known in the majority but not in all cases. Similarly, in many patients, the information about the exact tumour size was missing. Annual incomes were also provided as interval values in the SEER database. Therefore, we decided to keep the presented multivariable model in the current way to demonstrate the value of adequate LND and retain a large sample size for that analysis.

Based on your kind advice we added additional results showing the prognostic role of LN yield (as a continuous variable) for CSS using Cox proportional hazards regression adjusted for confounders.

“Additionally an independent association between the number of removed LNs (HR=0.96 95%CI 0.93-0.98 P<.01) and CSS was confirmed after adjustment for above factors in multivariable analysis.”

- all around the manuscript, p-values with 3-sign digits

- all around the manuscript, HRs and 95% CIs with 2-sign digits

Author Response: Corrections were made to unify data presentation and readability. In the manuscript, HR and 95% CI are presented with two decimal places, and p-values are presented with three decimal places. Thank you.

- Propensity-score matched cohort, all info is lacking, please provide full detail about this procedure (how have you done it!?)

- as a further step, please describe how you have used PS matching in the Cox PH models, actually this detail is lacking

Author Response: Thank you for this important notice. We added additional paragraph to the methods section to clarify the procedure of propensity score matching.

“Propensity score matching (PSM) was performed between patients who underwent LND and those who did not, with adjustment for the following confounders: T stage, grade, tumor size, tumor location, age, and gender. Additionally, forced matching was employed to ensure that treated ones and their matched control counterparts shared identical values for tumor T stage and tumor location. The propensity scores were calculated based on logistic regression to estimate the probability of receiving treatment, and the support region was extended by the logit of the propensity score. The method employed was optimal matching with a two-to-one ratio of control to treated to units in order to minimize the total within-pair difference, thereby enhancing the comparability of the study groups.”

We conducted propensity score matching to balance the LND and no LND cohorts based on the specified factors. As demonstrated in Table 3, the matched groups exhibited excellent balance and showed no significant differences in any of the measured confounding variables, including stage, grade, tumor size, location, chemotherapy, age, and gender. Subsequently, in this well-balanced cohort (N=2526), we performed Cox proportional hazard regression with a stepwise variable selection method using a significance level of 0.2 to enter and 0.05 to stay in the model.

Additional information was added to the methods section:

“Multivariable analysis was performed to identify independent predictors of CSM in the whole cohort and in the well-balanced cohort after PSM. The stepwise variable selection method was used with a significance level of 0.2 to enter and 0.05 to stay in the multivariable model for each variable. Additionally, competing-risk regression using CPH was performed to account for other-cause mortality.”

- please, add the number of OS and CSS events, globally and stratified by lymph node dissection 

Author Response: Thank you for this kind notice. Detailed numbers of events were previously presented in the Table 1 for the whole cohort and in the Table 2 for LND and no LND groups. We added additional information about these statistics into the results section:

In the LND group, there were 1205 (32.2%) cancer-specific deaths and 2136 (57.1%) all-cause deaths, whereas in the no LND group, there were 225 (24.8%) cancer-specific deaths and 398 (43.8%) all-cause deaths (table 2).

Information about OS and CSS events has already been available in the whole population:

“During the median study follow-up of 100 months (IQR 61-143 months), 2534 (54.5%) deaths were recorded and 1430 (30.8%) were attributable to UTUC.“

Reviewer 2 Report

Dear Authors

Thank you for your manuscript submission. This work is well-designed, interesting and effective. A Minor Revision is needed as below:

1. Lines 105 and 106: "Tumours located in the renal pelvicalyceal system were predominant (N=3038; 65.4%), ... "; (N=3038; 65.3%) is correct

Please do check all the mentioned percentages throughout the manuscript.

2. Please do read and add the following papers to References section of the manuscript to have a fruitful Discussion section:

Effect of lymph node dissection on stage-specific survival in patients with upper urinary tract urothelial carcinoma treated with nephroureterectomy. BMC cancer. 2019 Dec;19:1-0.   The Value of Lymph Node Dissection in Patients With Node-Positive Upper Urinary Tract Urothelial Cancer: A Retrospective Cohort Study. Frontiers in Oncology. 2022 Jun 16;12:889144.   Lymph node dissection during radical nephro-ureterectomy for upper tract urothelial carcinoma: a review. Frontiers in Surgery. 2022 Mar   Benefit of lymph node dissection in cN+ patients in the treatment of upper tract urothelial carcinoma: Analysis of NCDB registry. InUrologic Oncology: Seminars and Original Investigations 2022 Sep 1 (Vol. 40, No. 9, pp. 409-e9). Elsevier.   3. This study, deserves a stronger and more effective Conclusion section.

Author Response

Thank you for your manuscript submission. This work is well-designed, interesting and effective. A Minor Revision is needed as below:

Author Response: Thank you. We are honored for the kind reception of our work and greatly appreciate your suggestions for further analyses to improve our study.

  1. Lines 105 and 106: "Tumours located in the renal pelvicalyceal system were predominant (N=3038; 65.4%), ... "; (N=3038; 65.3%) is correct

Please do check all the mentioned percentages throughout the manuscript.

Author Response: Thank you for your notice. We have corrected and checked all other percentages mentioned in the text.

  1. Please do read and add the following papers to References section of the manuscript to have a fruitful Discussion section:

Effect of lymph node dissection on stage-specific survival in patients with upper urinary tract urothelial carcinoma treated with nephroureterectomy. BMC cancer. 2019 Dec;19:1-0.  

The Value of Lymph Node Dissection in Patients With Node-Positive Upper Urinary Tract Urothelial Cancer: A Retrospective Cohort Study. Frontiers in Oncology. 2022 Jun 16;12:889144.  

Lymph node dissection during radical nephro-ureterectomy for upper tract urothelial carcinoma: a review. Frontiers in Surgery. 2022 Mar  

Benefit of lymph node dissection in cN+ patients in the treatment of upper tract urothelial carcinoma: Analysis of NCDB registry. InUrologic Oncology: Seminars and Original Investigations 2022 Sep 1 (Vol. 40, No. 9, pp. 409-e9). Elsevier.   3.

Author Response: Thank you for this thoughtful comment. All citations were added and contributed to the improvement of the discussion section. The following sentences were added in different parts of the discussion.

“Consequently, the removal of more LNs may be associated with the eradication of unnoticed LN micrometastases and a subsequent improvement in the recurrence rate and CSS, suggesting that an extended template-based lymphadenectomy may have therapeutic benefits (17).”

“Zhai et al. reported the beneficial effect of LND, especially when 4 or more regional lymph nodes were removed, on survival in pT3/4 UTUC (18). But, it is worth noting that this analysis included a proportion of node-positive patients (18).”

“Noteworthy, the survival benefit from LND in node-positive UTUC is also controversial and probably limited to selected patients with low burden of LN metastatis, in whom LND might be therapeutic. Hao-Ran et al. demonstrated that in patients with node-positive UTUC, removing more LNs does not offer a better therapeutic effect. However, positive lymph node density provided additional prognostic value for CSS and OS (21). Another study showed that in clinically node-positive patients with UTUC, performing LND in addition to RNU at any clinical stage does not seem to have a significant impact on OS (22).”

This study, deserves a stronger and more effective Conclusion section.

 Author Response: Thank you for this thoughtful comment. After careful consideration, we have revised and improved our conclusion based on your insights. The modified conclusion now more effectively summarizes our findings and their implications.

“In conclusion, our population-based study demonstrated that adequate LND during RNU is associated with improved overall and cancer-specific survival in patients with lymph node-negative UTUC, particularly in individuals with muscle-invasive disease. However, LND remains underutilized during RNU and most often resulting in low lymph node yield. LND provides essential staging information and might lead to eradication of undetected nodal micrometastasis, ultimately contributing to improved oncologic outcomes. Further prospective trials are warranted to validate our findings.“

Reviewer 3 Report

General comment

The manuscript entitled “The association between lymph node dissection and survival in lymph node-negative upper urinary tract urothelial cancer.” By Åšlusarczyk et al., aims to evaluate the association between LND and its extend and survival in LN-negative UTUC, using the data retrieved from the SEER database. Despite data are used from an already known and widely analyzed database, the manuscript is well written and built, reporting an interesting topic in an clear and concise manner. Few minor corrections are suggested.

INTRODUCTION

47: check grammar.

49: you could be more precise in this case.

RESULTS

116-119: this could be also the reason for better improved survival in patients undergoing LND.

DISCUSSION

179: consider this paper in your discussion DOI: 10.1016/j.clgc.2023.08.001

267: propose future research and possible changes in the clinical practice.

CONCLUSIONS

279: The conclusion could be polished in terms of delivery of the final findings.

Minor grammar checks

Author Response

General comment

The manuscript entitled “The association between lymph node dissection and survival in lymph node-negative upper urinary tract urothelial cancer.” By Åšlusarczyk et al., aims to evaluate the association between LND and its extend and survival in LN-negative UTUC, using the data retrieved from the SEER database. Despite data are used from an already known and widely analyzed database, the manuscript is well written and built, reporting an interesting topic in an clear and concise manner. Few minor corrections are suggested.

Author Response: Thank you. We are honored for the kind reception of our work and greatly appreciate your suggestions for further analyses to improve our study.

INTRODUCTION

47: check grammar.

49: you could be more precise in this case.

Author Response: We really appreciate your thoughtful notice. We added additional sentence to clarify that issue and we modified the first sentence: “The oncological benefit of lymph node dissection (LND) during RNU remains controversial as well-designed randomized trials are lacking (2,3). A systematic review of nine retrospective studies demonstrated improved cancer-specific survival (CSS) and reduced risk of local recurrence when complete and template-based LND is performed in patients with high-stage UTUC (≥pT2).”

RESULTS

116-119: this could be also the reason for better improved survival in patients undergoing LND.

Author Response: Thank you for this important notice

DISCUSSION

179: consider this paper in your discussion DOI: 10.1016/j.clgc.2023.08.001

267: propose future research and possible changes in the clinical practice.

Author Response: Thank you for these thoughtful suggestions. Above paper was cited and additional sentences were added to the discussion section.

“Several studies have reported an independent negative prognostic role of older age for CSS in various stages of urothelial cancer of the bladder and upper urinary tract (22–25). A recent multicenter study has also shown that patients undergoing RNU for UTUC aged 70 and above had worse OS and CSS compared to younger individuals (25).”

Additional sentences were added to the discussion to underline the need for future research and suggest possible changes in the clinical practice:

“Only a meticulously designed randomized trial can definitively address the question of the benefits and optimal extent of LND in patients with UTUC.”

“Based on available evidence and our observations, a template-based LND should be an inherent part of RNU for muscle-invasive UTUC, aiming for more adequate staging and potential eradication of regional metastasis.”

CONCLUSIONS

279: The conclusion could be polished in terms of delivery of the final findings.

Author Response: Thank you for this thoughtful comment. We greatly appreciate your feedback and constructive suggestions. After careful consideration, we have revised and improved our conclusions based on your insights. The modified conclusion section now more effectively summarizes our findings and their implications.

“In conclusion, our population-based study demonstrated that adequate LND during RNU is associated with improved overall and cancer-specific survival in patients with lymph node-negative UTUC, particularly in individuals with muscle-invasive disease. However, LND remains underutilized during RNU and most often resulting in low lymph node yield. LND provides essential staging information and might lead to eradication of undetected nodal micrometastasis, ultimately contributing to improved oncologic outcomes. Further prospective trials are warranted to validate our findings